**Data Availability Statement:** All relevant data are within the paper and its Supporting Information files.

**Funding:** NIH, NIAID: R01AI147347-01 "Identifying TB transmitters in high TB/HIV

# Sensitivity optimisation of tuberculosis bioaerosol sampling

**Benjamin Patterson**[1]*, **Ryan Dinkele**[2,3], **Sophia Gessner**[2,3], **Carl Morrow**[4], **Mireille Kamariza**[5], **Carolyn R. Bertozzi**[6,7], **Andrew Kamholz**[8], **Wayne Bryden**[9], **Charles Call**[9], **Digby F. Warner**[2,3,10], **Robin Wood**[3,4]

**1** Amsterdam Institute for Global Health and Development, University of Amsterdam, Amsterdam, Netherlands, **2** Department of Pathology, Faculty of Health Sciences, SAMRC/NHLS/UCT Molecular Mycobacteriology Research Unit & DST/NRF Centre of Excellence for Biomedical TB Research, University of Cape Town, South Africa, **3** Institute of Infectious Disease and Molecular Medicine, Faculty of Health Sciences, University of Cape Town, Cape Town, South Africa, **4** Desmond Tutu HIV Centre, University of Cape Town, Cape Town, South Africa, **5** Department of Biology, Stanford University, Stanford, California, United States of America, **6** Department of Chemistry, Stanford University, Stanford, California, United States of America, **7** Howard Hughes Medical Institute, Stanford University, Stanford, California, United States of America, **8** Edge Embossing, Charlestown, Boston, Massachusetts, United States of America, **9** Zeteo Tech, Sykesville, Maryland, United States of America, **10** Wellcome Centre for Infectious Disease Research in Africa, University of Cape Town, Cape Town, South Africa

* patterson.b@unic.ac.cy

## Abstract

### Introduction

Detection of *Mycobacterium tuberculosis* (*Mtb*) in patient-derived bioaerosol is a potential tool to measure source case infectiousness. However, current bioaerosol sampling approaches have reported low detection yields in sputum-positive TB cases. To increase the utility of bioaerosol sampling, we present advances in bioaerosol collection and *Mtb* identification that improve detection yields.

### Methods

A previously described Respiratory Aerosol Sampling Chamber (RASC) protocol, or "RASC-1", was modified to incorporate liquid collection of bioaerosol using a high-flow wet-walled cyclone (RASC-2). Individuals with GeneXpert-positive pulmonary TB were sampled pre-treatment over 60-minutes. Putative *Mtb* bacilli were detected in collected fluid by fluorescence microscopy utilising DMN-Trehalose. Exhaled air and bioaerosol volumes were estimated using continuous $CO_2$ monitoring and airborne particle counting, respectively. *Mtb* capture was calculated per exhaled air volume sampled and bioaerosol volume for RASC-1 (n = 35) and for RASC-2 (n = 21). Empty chamber samples were collected between patients as controls.

### Results

The optimised RASC-2 protocol sampled a median of 258.4L (IQR: 226.9–273.6) of exhaled air per patient compared with 27.5L (IQR: 23.6–30.3) for RASC-1 (p<0.0001). Bioaerosol volume collection was estimated at 2.3nL (IQR: 1.1–3.6) for RASC-2 compared with 0.08nL

burdened communities" (Robin Wood, PI; Digby Warner, co-PI). 1st July 2019 – 30th June 2024 the South African Medical Research Council (MRC) with funds from National Treasury under the Economic Competitiveness and Support Package (MRC-RFA-UFSP-01- 2013/CCAMP) and the Bill & Melinda Gates Foundation (OPP1116641). (Robin Wood, PI) The funders provided support in the form of salaries for authors RW and DW, but did not have any additional role in the study design, data collection and analysis, decision to publish, or preparation of the manuscript.

**Competing interests:** WB and CC are employed by Zeteo Tech. AK is employed by Edge Embossing. Belonging to these commercial entities does not alter the authors' adherence to PLOS ONE policies on sharing data and materials.

(IQR: 0.05–0.10) for RASC-1 ($p<0.0001$). The detection yield of viable *Mtb* improved from 43% (median 2 CFU, range: 1–14) to 95% (median 20.5 DMN-Trehalose positive bacilli, range: 2–155). These improvements represent a lowering of the limit of detection in the RASC-2 platform to 0.9 *Mtb* bacilli per 100L of exhaled air from 3.3 *Mtb* bacilli per 100L (RASC-1).

## Conclusion

This study demonstrates that technical improvements in particle collection together with sensitive detection enable rapid quantitation of viable *Mtb* in bioaerosols of sputum positive TB cases. Increased sampling sensitivity may allow future TB transmission studies to be extended to sputum-negative and subclinical individuals, and suggests the potential utility of bioaerosol measurement for rapid intervention in other airborne infectious diseases.

## Introduction

Tuberculosis (TB) remains a serious global public health threat [1]. In 2018, the global incidence was estimated at 10 million with a death toll of 1.5 million [2]. Driving the high burden is on-going transmission of infection, as demonstrated by the 1.1 million cases of childhood TB disease globally. Further evidence for the prominence of transmission comes from molecular epidemiology studies in high-prevalence countries which have highlighted that the majority of TB disease follows from recent *Mtb* infection [3].

Transmission events are influenced by many factors which relate to the source case, the bacillus, the environment and the host. Fundamental to transmission, however, is the production of aerosolized *Mtb*: in the absence of aerosol release, no transfer of infectious organisms between individuals can occur. Pioneering experiments performed by Wells and Riley more than 60 years ago utilized the guinea pig as an aerosol sampler, detecting transmission by the demonstration of new infection in the animals [4,5]. Employing a sampling system with physical separation of patients (source case) and guinea pigs (sampler) led to the confirmation of airborne transmission. Moreover, intricate linking of individual patients to specific guinea pig infections by bacillary resistance patterns and timing of entry and departure from the ward allowed for identification of a minority of highly infectious 'super-spreaders'. The guinea pig model therefore supported the concepts that infectiousness is highly heterogenous and that small bioaerosol–which arise as a consequence of dehydration of buoyant airborne droplets during "aging" following release into the environment–are primarily responsible for remote transmission [6].

Invaluable though this contribution was to the early understanding of TB transmission, the experimental design was not optimised for sensitive sampling [7]. Minimal sensitivity to *Mtb* production necessitates sampling multiple patients for days to months for both historical and recent *in vivo* sampling experiments [8–10]. However, some modern approaches with greater proximity between source case and sampler, single patient sampling and viable organism detection have dramatically improved the overall detection sensitivity. Direct detection has identified *Mtb* bacilli within only minutes to hours of sampling [11–13], generated individual-level information, and found a greater proportion of emitters.

These sensitive techniques have the potential to efficiently answer questions regarding the transmission potential of patient sub-groups and test strategies for transmission mitigation.

Further sensitivity improvement may allow investigation of *Mtb*-containing droplet release from patients across the TB disease spectrum, including sputum smear-negative cases, HIV-coinfected and even subclinical TB cases. Sensitive sampling combined with quantitative forms of detection also provide an infectivity measure. Such a measure could be applied longitudinally to those receiving chemotherapy to more accurately establish time to bioaerosol sterilisation with direct clinical application.

We describe a direct bioaerosol sampling device designed to collect aged aerosol. We have attempted to optimise key components necessary to maximise detection yield through an iterative re-design process. Modifications to both the sampling process and the method of organism detection are described with the results of sampling from both healthy controls and individuals with newly diagnosed pulmonary TB presented to evaluate the improvements.

## Materials and methods

The Respiratory Aerosol Sampling Chamber (RASC) is an enclosed, clean space optimised for the sampling of aged bioaerosol from a single individual during a 60-minute period of natural respiratory activity. A previous description of the RASC design, "RASC-1", has been published [14] in addition to results from a 35-patient pulmonary TB sampling study [11]. Briefly, the RASC is a 1.4m$^3$ chamber in which participants can comfortably sit throughout the study period. The chamber is first sealed and then an air purge phase is performed by drawing external air across HEPA filters for a 10-minute period. The next phase is passive contamination with respiratory bioaerosol as the participant respires. A sampling phase then occurs with various devices drawing volumes of air out of the chamber and extracting the airborne bioaerosol. Finally, a second 10-minute purge phase is performed to remove residual *Mtb* bioaerosol.

The apparatus and sampling protocols employed for those initial studies is referred to throughout this report as "RASC-1". The current study describes several design modifications, informed by an iterative series of experiments, which have refined the apparatus and sampling protocol while maintaining the basic structure of the chamber. This new version is referred to as "RASC-2".

Table 1 summarises the improvements in the sampling and detection systems which are further explained in the following text.

### Sample collection

RASC-1 included an array of aerosol sample collection instruments operated consecutively at the peak of chamber air contamination during the sampling phase. This included polycarbonate, gelatin and felt filters, Dekati and Andersen impactors, and a cyclone collector. The flow rates (chamber outflow) varied between instruments, subject to device specifications. Both

**Table 1. Comparison of the features modified from RASC-1 to RASC-2.**

| Optimisation Parameter | RASC-1 | RASC-2 | Advantage |
|---|---|---|---|
| **Sampler** | Primarily the Andersen impactor | Cyclone collector (Coriolis μ air sampler) | Liquid collection may improve in recovery of live *Mtb* bacilli |
| **Sampling Rate and Time Period (see Fig 1)** | 28L/min for 10 minutes | 250L/min for 60 minutes | Increased exhaled air volume (estimated by $CO_2$. See Fig 2A.) |
| | | | Increased bioaerosol content (estimated by aerodynamic particle counting. See Fig 2B). |
| | | | Improved regulation of humidity and temperature |
| **Detection Method** | *Mtb* Culture (Middlebrook 7H10 Solid Agar) | DMN-trehalose dye | More rapid result |
| | | | Retains high sensitivity (see Fig 4) Indication of viability via requirement for metabolic activity for DMN-tre dye uptake |

impactors and the polycarbonate and gelatin filters sampled with flow rates of 20–30 L/min, whereas the felt filter and the cyclone collector sampled with higher flow rates: 300 and 250 L/min, respectively. Results yielded no positive samples from the felt or gelatin filters. For the rest, no sampling instruments demonstrated significantly higher yield per unit of exhaled air sampled. Therefore, to maximize the exhaled air volume sampled, the sample collection process was simplified to a single cyclone collector for RASC-2 (described below).

### Cyclone collector

In RASC-2 a high-efficiency cyclone collector (Coriolis μ air sampler, Bertin, Montigny le Bretonneux, France) was used to collect airborne bioaerosol into 5–10 mL of sterile phosphate-buffered saline (PBS) solution. Mounted within the chamber, the air enters the conical collector through a tangential nozzle. The high flow rate of 250 L/min generates a liquid cyclone and particle inertia leads to deposition of bioaerosol from the airstream onto the wet wall with a high collection efficiency. Collection into liquid has the advantage of minimizing particle bounce, improving the recovery of viable *Mtb* bacilli and maximizing the air volume sampled over time. (see appendix for schematic diagrams of RASC-1 (modified from ref 14.) and RASC-2).

### Sampling protocol

The experimental protocol was modified to continuous sampling of the chamber air via the single high-flow (250 L/min) cyclone collector. This replaces the sequential sampling phases of the RASC-1 protocol. Consequently, there is a lowering of the $CO_2$ set-point throughout the protocol and improved regulation of chamber humidity and temperature, illustrated by the $CO_2$ traces in Fig 1. This maximizes the air volume sampled by reducing the loss of exhaled air at the end of the study protocol. There is an added benefit of improved thermal comfort for participants.

### *Mtb* detection

Liquid specimen collection facilitates sample concentration via centrifugation and the application of culture-independent detection techniques. For the RASC-2 protocol, the centrifuged bioaerosol sample pellet was incubated with a novel solvatochromic trehalose analog, DMN-Trehalose, which was incorporated into the cell walls of metabolically active mycobacteria, and subsequently yielding high fluorescence intensity detectable by a fluorescence microscope [15]. Each participant sample was added to a microwell device (Edge Technologies, USA) containing an array of 1600 nanowells measuring 50 x 50 μm. This generated multiple discrete microenvironments for *Mtb* detection and avoided overgrowth of contaminants affecting the whole sample [16]. Fluorescent microscopy employed a skilled operator to distinguish *Mtb* on the basis of bacillus morphology and specific staining patterns.

### $CO_2$ and bioaerosol monitoring

Continuous $CO_2$ and aerodynamic particle monitoring of the chamber air was performed in both sampling protocols, RASC-1 and RASC-2. Estimates of captured *Mtb* bacilli per unit volume of exhaled air and per unit volume of respiratory bioaerosol were calculated using the method described in the previous (RASC-1) study [12].

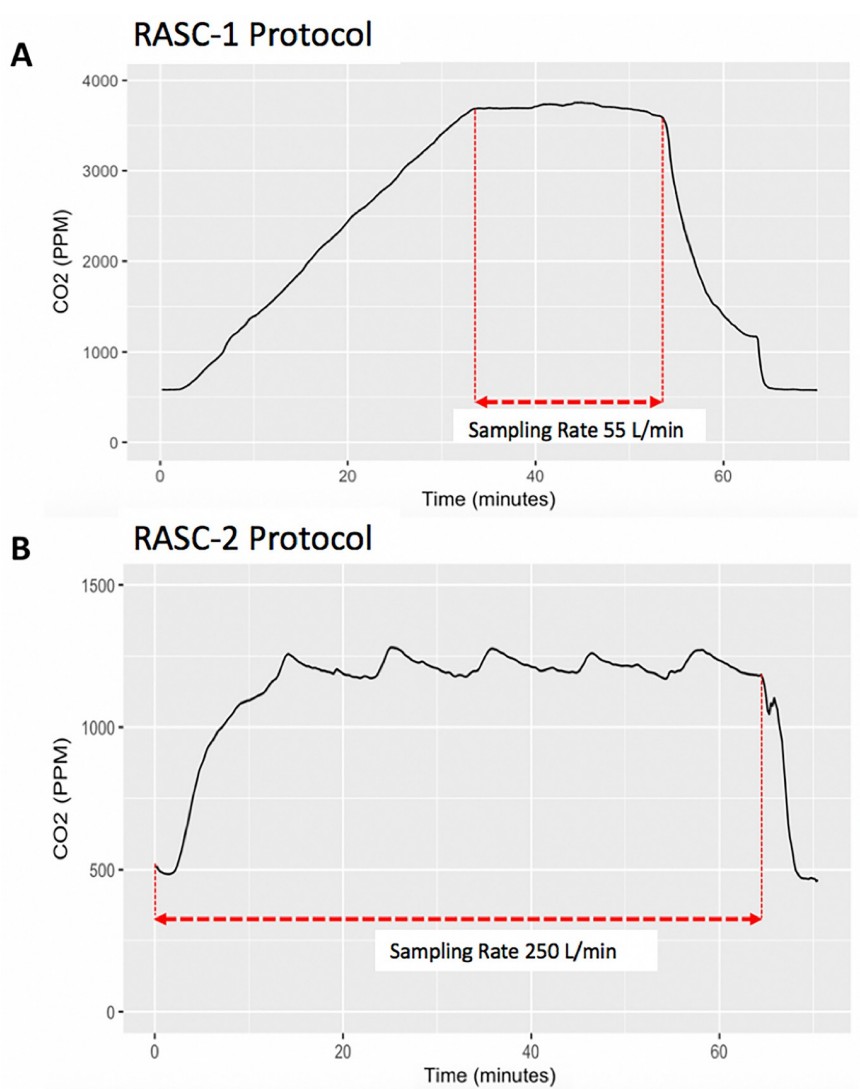

**Fig 1.** Differing example $CO_2$ traces for the two sampling protocols (A) RASC-1: Initial contamination and short, steady-state sampling (median 3800 PPM $CO_2$). (B) RASC-2: Continuous sampling with a lower steady-state (median 1600 PPM $CO_2$).

## Study participants

All recruited participants were residents from two informal peri-urban settlements outside Cape Town and in close proximity to the study site. Eligible candidates were those with GeneXpert-positive sputum without evidence of drug resistance. Bioaerosol sampling was performed before initiation of anti-tuberculous therapy, as per the approved Ethics protocol (see below). The empty chamber was sampled between human participants to serve as chamber decontamination controls. Baseline patient data were collected from the clinical records and a chest X-ray was taken approximately seven days after the start of treatment. The presence of lung cavitation was scored by one of the authors (BP) based on the chest X-ray and this score was compared to a radiologist report for agreement.

### Statistical analysis

The performance of the RASC-2 protocol was compared with the original RASC-1 design in terms of the exhaled air volume and bioaerosol volume sampled and *Mtb* bacilli captured. Comparison between the protocols was made with a Wilcoxon rank sum test. Statistical analyses were performed using R Core Team (2019).

### Ethical statement

The patient studies are covered by two separate ethics approvals from the University of Cape Town Faculty of Health Sciences Human Research Ethics Committee (HREC/REF: 680/2013; HREC/REF: 529/2019). Each patient provided written informed consent prior to participation in the study, including consent to publication of the clinical and demographic details.

## Results

### Patient characteristics

Individuals were consecutively recruited from the same two communities. The RASC-1 protocol was implemented over the time period 2015 to 2017 and RASC-2 from 2018 to 2019. Table 2 compares basic clinical and demographic characteristics, confirming broad equivalence between patients sampled by the two protocols.

### RASC comparison

The two RASC designs were compared using indicators of sampling efficiency (Fig 2A and 2B) and numbers of *Mtb* bacilli captured (Fig 2C). The sampled volume of exhaled air was calculated using continuous $CO_2$ measurement to determine exhaled air proportion in chamber air during device sampling. Similarly, chamber bioaerosol concentration was determined by particle counting and volume calculation with the assumption that bioaerosol are spherical. Captured bioaerosol volume was estimated by assuming completely efficient sampling of this concentration. Both these methods have been described previously [11].

**Table 2. Baseline demographics and clinical characteristics of RASC-1 protocol and RASC-2 protocol.**

|  | RASC -1 | RASC -2 | p-value |
|---|---|---|---|
| **N** | 35 | 21 | |
| **Age** | | | |
|   median [IQR] | 32 [25,39] | 35 [31,41] | 0.23 |
| **Sex** | | | |
|   Male (%) | 20 (57.1) | 14 (77.8) | 0.24 |
| **HIV Status** | | | |
|   Positive (%) | 17 (48.6) | 6 (28.6) | 0.23 |
|   CD4+ Count | | | |
|     median [IQR] | 122 [63,417] | 90 [49,146] | 0.51 |
| **Sputum GeneXpert** | | | |
|   Positive (%) | 35 (100) | 21 (100) | N/A |
| **Previous TB** | | | |
|   Yes (%) | 11 (31.4) | 5 (23.8) | 0.76 |
| **Cavitation**[*] | | | |
|   Present (%) | 13 (39.4) | 6 (46.2) | 0.93 |

[*]Chest radiograph performed in 33 patients in RASC-1 and in 12 patients from the RASC-2 group

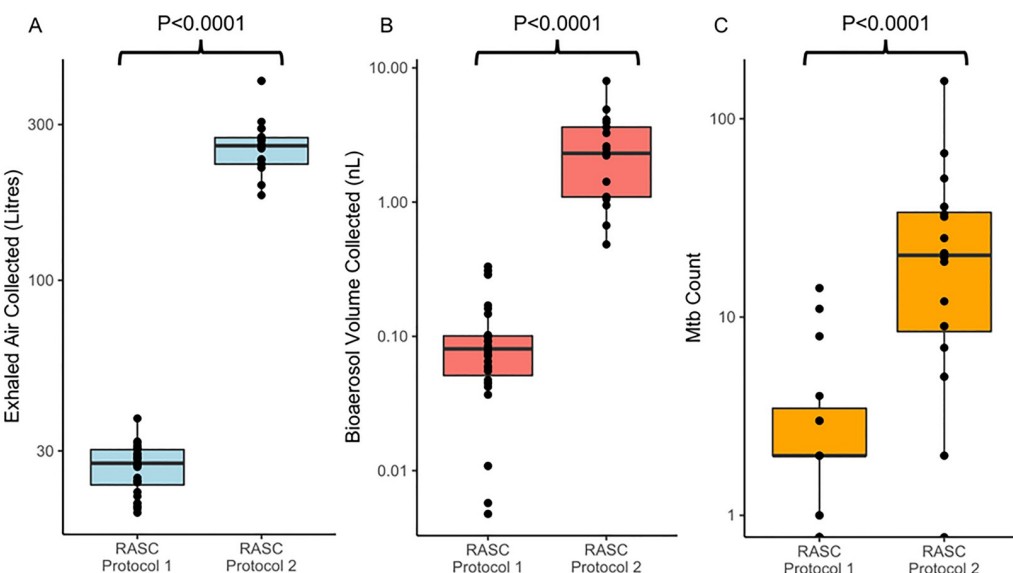

**Fig 2. Comparative efficiencies of RASC-1 versus RASC-2 in capture of exhaled air (A), collection of bioaerosol volume (B), and capture of *Mtb* organisms (C).** Note that numbers in (C) were determined by colony forming units (RASC-1) and DMN-Trehalose-positive organisms identified using fluorescence microscopy (RASC-2).

## RASC decontamination

As mentioned previously [14], the highly sensitive *Mtb* detection methodology used in RASC-2 is vulnerable to contamination owing to incomplete sterilization of the chamber. To control for this risk, patient sampling was interspersed with empty RASC sampling. Detected organisms in the control samples suggested a small degree of contamination. Fig 3 demonstrates the different bacillary counts for patient sampling and empty RASC sampling. Control samples remained negative after modification of the collector design and high concentration ozone sterilization of the RASC.

## Limit of detection

A bacillary count per unit volume of exhaled air was calculated and compared between the two sampling protocols. Fig 4 illustrates the range of detectable concentrations across the sampled populations. Notably, by combining the design improvements, the limit of detection has been lowered from approximately 3.3 bacilli per 100 L exhaled air to 0.9 bacillus per 100 L.

## Discussion

This study has shown that, by optimising both bioaerosol collection and *Mtb* detection, a majority of newly diagnosed GeneXpert sputum positive TB patients can be shown to exhale viable *Mtb* organisms. The increased numbers of *Mtb* organisms captured by RASC-2 compared with RASC-1 were associated with an approximate 1 $\log_{10}$ increase in exhaled air sampled and bioaerosol particle volume collected. This is not explained by demographic and clinical characteristics which were broadly equivalent between the patients sampled by the different methods. The resulting lowered limit of detection enabled identification of putative *Mtb* organisms in 95% of patients whose bacillary concentrations varied over a 2 $\log_{10}$ range from 1 to 100 per 100 L exhaled air. The proportion of patients identified with *Mtb* positive bioaerosols is therefore a function of lower level of detection. However, exhaled particle volumes

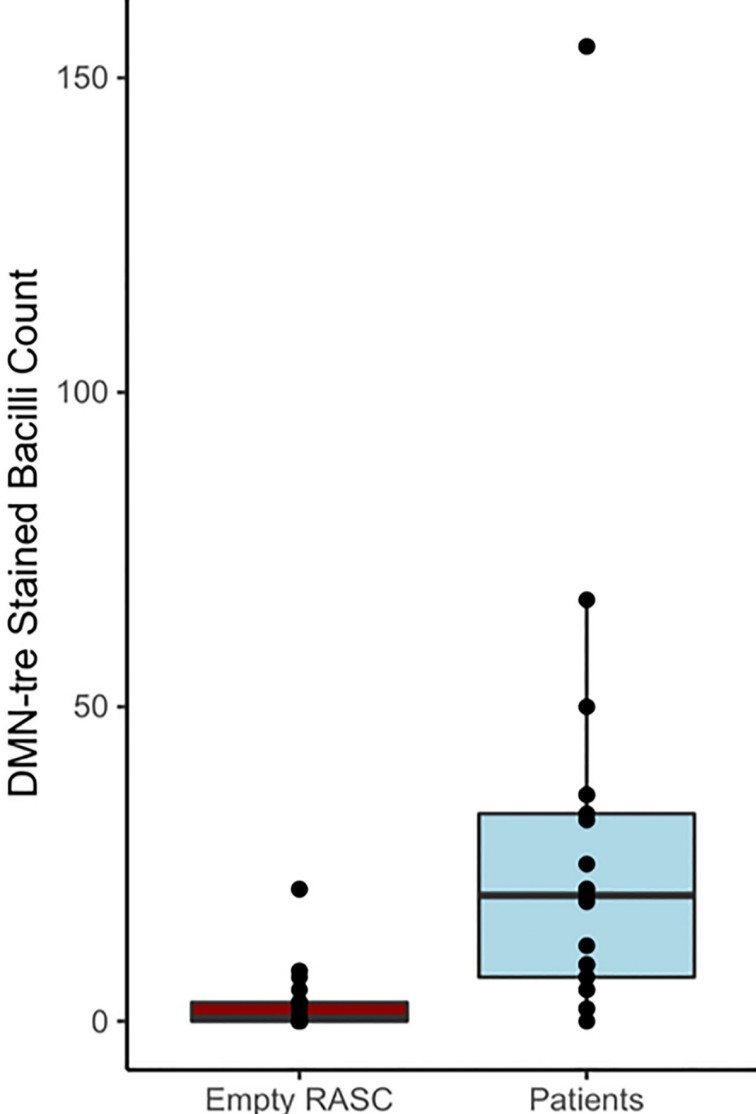

**Fig 3. Comparison of total bacillary counts following RASC-2 sampling of sputum GeneXpert-positive TB patients (n = 21) and the inter-patient empty booth RASC sampling controls (n = 22).** P<0.0001.

varied markedly between individuals, suggesting that infectivity is likely to be impacted by both *Mtb* concentration of lung fluids and individual bioaerosol particle production.

The current study sampled newly diagnosed TB patients while seated in a small personalized clean room with normal respiration and spontaneous coughing over 60-minutes. Collected bioaerosol were those which remained airborne in the RASC and, at time of collection, were predominantly in the 1–5 μm diameter range. Consistent with prior observations [6], these are the particles thought most likely to reach the peripheral lungs of susceptible individuals sharing an indoor location [17]. It is also probable that these bioaerosols containing viable *Mtb* organisms might be indicative of a wide range of source infectivity. The investigation of *Mtb* production and, consequently, potential infectivity during various respiratory manoeuvres is under current investigation in our laboratory.

The sensitive detection of individual *Mtb* organisms, with DMN-Trehalose staining and subsequent fluorescence microscopy analysis, requires a high degree of sterility to exclude

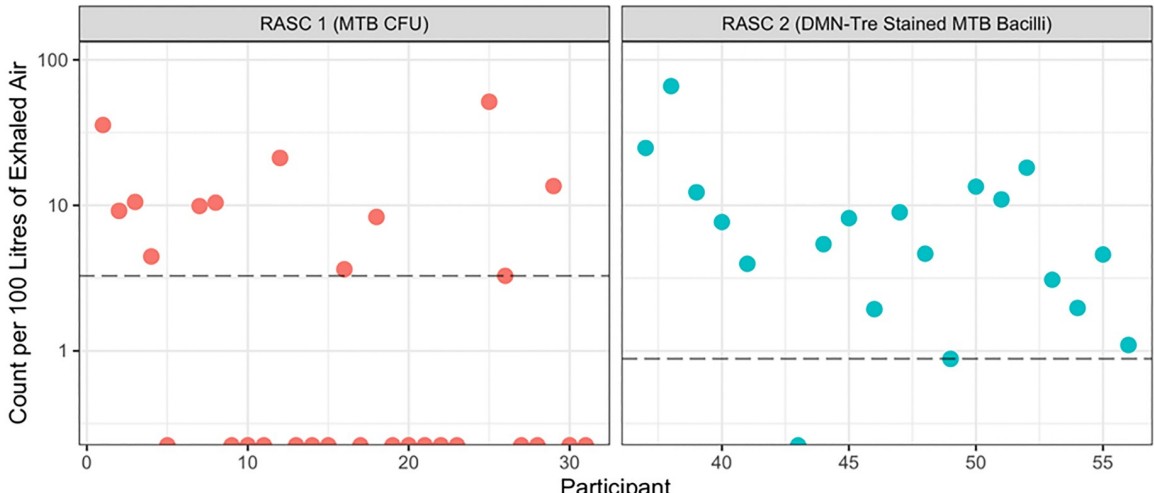

**Fig 4. *Mtb* bacillary counts per 100 L exhaled air for RASC-1 versus RASC-2.** Sampled individuals not yielding detectable *Mtb* bacilli are illustrated by points on the x-axis.

contamination during sampling and isolation. After detailed analysis of the platform and collection equipment, we noted that the wet-walled cyclone contained a dead space which was a potential reservoir for live *Mtb* cells. Therefore, the device was modified to enable dismantling and autoclaving. The very low numbers of *Mtb* organisms detected during empty booth sampling could also have resulted from the release of trapped *Mtb* organisms by the negative pressure inducing retrograde airflow across the exit HEPA filter. Reassuringly, following modification of the collector and the application of combined dry vapour hydrogen peroxide and ozone treatments of the RASC after use, the empty booth controls remained negative. A limitation of this study is that modifications to the sampling protocol and the detection method were evaluated in combination. It is therefore not possible to delineate the extent to which individual improvements impact sensitivity.

## Conclusion

The development of a sensitive and reproducible measure of *Mtb* in exhaled bioaerosols has enabled the demonstration of a wide range of potential infectivity of newly diagnosed TB patients. Bioaerosol assays may offer a useful adjunct to TB transmission studies, enabling demonstration of infectivity of sputum negative individuals and the effect of specific chemotherapies on infectivity. Moreover, the protocols and equipment described here suggest the potential application of this technology to other airborne infectious diseases, especially where rapid interventions such as containment and quarantine are required to curb outbreaks [18].

## Supporting information

**S1 Appendix.**
(TIF)

## Author Contributions

**Conceptualization:** Benjamin Patterson, Carl Morrow, Wayne Bryden, Charles Call, Digby F. Warner, Robin Wood.

**Data curation:** Benjamin Patterson.

**Formal analysis:** Benjamin Patterson, Robin Wood.

**Funding acquisition:** Digby F. Warner, Robin Wood.

**Investigation:** Ryan Dinkele, Sophia Gessner.

**Methodology:** Carl Morrow, Mireille Kamariza, Carolyn R. Bertozzi, Wayne Bryden, Charles Call, Digby F. Warner, Robin Wood.

**Project administration:** Ryan Dinkele, Sophia Gessner, Robin Wood.

**Resources:** Carolyn R. Bertozzi, Andrew Kamholz, Wayne Bryden, Charles Call.

**Supervision:** Digby F. Warner, Robin Wood.

**Validation:** Robin Wood.

**Visualization:** Benjamin Patterson, Robin Wood.

**Writing – original draft:** Benjamin Patterson, Robin Wood.

**Writing – review & editing:** Sophia Gessner, Carl Morrow, Mireille Kamariza, Digby F. Warner.

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
