## [Decision Letter · Decision Letter 0]

11 Jun 2020

PONE-D-20-11317

Sensitivity Optimisation of Tuberculosis Bioaerosol Sampling

PLOS ONE

Dear Dr. Patterson,

Thank you for submitting your manuscript to PLOS ONE. After careful consideration, we feel that it has merit but does not fully meet PLOS ONE’s publication criteria as it currently stands. Therefore, we invite you to submit a revised version of the manuscript that addresses the points raised during the review process.

We look forward to receiving your revised manuscript.

Kind regards,

Selvakumar Subbian, Ph.D.

Academic Editor

PLOS ONE

Journal Requirements:

The authors have declared that no competing interests exist.

We note that one or more of the authors are employed by a commercial company: Edge Embossing and Zeteo Tech.

3. Please include your tables as part of your main manuscript and remove the individual files. Please note that supplementary tables be uploaded as separate "supporting information" files.

4. Please include a caption for figure 1.

Reviewers' comments:

Reviewer's Responses to Questions

**Comments to the Author**

1. Is the manuscript technically sound, and do the data support the conclusions?

Reviewer #1: No

Reviewer #2: Partly

Reviewer #3: Yes

Reviewer #4: Yes

2. Has the statistical analysis been performed appropriately and rigorously? 

Reviewer #1: I Don't Know

Reviewer #2: No

Reviewer #3: Yes

Reviewer #4: Yes

3. Have the authors made all data underlying the findings in their manuscript fully available?

Reviewer #1: No

Reviewer #2: Yes

Reviewer #3: Yes

Reviewer #4: Yes

4. Is the manuscript presented in an intelligible fashion and written in standard English?

Reviewer #1: Yes

Reviewer #2: Yes

Reviewer #3: Yes

Reviewer #4: Yes

5. Review Comments to the Author

Reviewer #1: The authors describe some changes to an aerosol sampling procedure that appears to improve the sensitivity in detecting sputum-positive TB patients who exhale infectious aerosols. There are some serious problems with the manuscript.

1) The paper compares data from the current study (RASC-2) with a previous study (RASC-1). Data from RASC-1 have already been published (Ref 14) and are repeated here.

2) The two studies cannot be compared. They were conducted 2-3 years apart with different patient groups. More importantly, two different measurements were used to determine the number of mycobacteria in the sampled aerosols – viable bacilli or cfu (RASC-1) and fluorescence microscopy (RASC-2). The authors provide no evidence that the fluorescently-labeled bacilli in RASC-2 were viable.

3) The authors admit (lines 143-144) that the fluorescence microscopy detection assay for detecting mycobacteria in RASC-2 was subjective and dependent upon the skill of the operator. Please provide some data or references to support the validity of this assay (e.g., repeated determinations from the same sample).

4) The overall conclusion of the study is trivial and can be summarized in a single sentence – “Increasing the amount of exhaled air sampled will increase the sensitivity of bacillary detection”. This is intuitively obvious to any intelligent person and does not require data or a manuscript to confirm.

5) The authors have grossly over interpreted their results. They speculate wildly about the possible implications of their aerosol sampling results with no supporting data.

6) The authors utilize jargon that is not defined for the reader. What, precisely, is an “aged” aerosol? What is the difference between “exhaled air” and “respiratory bioaerosol”?

Reviewer #2: In the present study, the authors have presented the method to sample bioaerosol from TB patients in order to determine the infectiousness. The method they described is an improvement of their previous work. However, I find following concerns,

1. Data for TB cases has been given for year 2018, instead of 2019.

2. The technical improvements in the design of RASC has not been elaborately explained. The focus has more been put on the collection outputs of aerosols in results section. I think, it will be better to present collection output in relation to modifications in the design of RASC.

3. In the method section, superficial information has been given. They should describe all the modifications in detail.

4. In the sample collection section, it is difficult to understand the technicalities. The authors should present these elaborately with proper diagrams and flowcharts.

5. In fig3, statistical significance and p value are missing.

6. Figure legends are not self explanatory, more details are required.

Reviewer #3: In the present study, authors employed modified version of Respiratory Aerosol Sampling Chamber (RASC) to incorporate liquid collection of bioaerosol using a high-flow wet-walled cyclone (RASC-2). Authors demonstrated this technical modification increased the utility of bioaerosol sampling and lowered the limit of Mtb detection in RASC-2 platform. Study result is interesting and has future application in characterizing exhaled air and bioaerosol from sputum negative and subclinical individuals. Authors have to address the following comments.

Major Comments

1. Authors estimated the Mtb count per 100 liters of exhaled air as Mtb-CFU for RASC and DMN-Tre stained Mtb bacilli. Since authors employed two different methodology to estimate the lower limit of detection, study result may vary. Please clarify this discrepancy in methodology.

2. This includes patients with and without cavitary TB. Earlier studies have shown that pulmonary TB patients with lung cavitation are the main source of disease transmission. Did you analyze the data such as Mtb count and lower limit of Mtb detection between the patients with and without cavitary TB in both RASC-1 and RASC-2.

3. Is it possible to study the nature of Mtb such as Mtb in the form of singles, small and large clumps in exhaled air and bioaerosol?

4. There is typo in the table 1; the number of patients with lung cavitation in RASC-2 is 6 (28.6%) not 46.2%. Correct the typo.

Reviewer #4: Authors have tried to improvise on a previously established protocol for aerosol sampling from the patients for possible bacterial count. The sensitivity can improvise the testing capability for possible MTB clearance.

6. PLOS authors have the option to publish the peer review history of their article (what does this mean?). If published, this will include your full peer review and any attached files.

Reviewer #1: No

Reviewer #2: No

Reviewer #3: No

Reviewer #4: No

---

## [Author Response · Author response to Decision Letter 0]

2 Jul 2020

Journal Requirements:

and

I have amended the document and file names and types in line with these requirements

The authors have declared that no competing interests exist.

We note that one or more of the authors are employed by a commercial company: Edge Embossing and Zeteo Tech.

A competing interests section has been added to the title page. 

A funding statement has been added to the title page. 

3. Please include your tables as part of your main manuscript and remove the individual files. Please note that supplementary tables be uploaded as separate "supporting information" files.

4. Please include a caption for figure 1.

This has been added to the manuscript (apologies all figure captions were not the latest version in the original submission – this has now been corrected)

Reviewers' comments:

Reviewer's Responses to Questions

Comments to the Author

1. Is the manuscript technically sound, and do the data support the conclusions?

Reviewer #1: No

Reviewer #2: Partly

Reviewer #3: Yes

Reviewer #4: Yes

2. Has the statistical analysis been performed appropriately and rigorously?

Reviewer #1: I Don't Know

Reviewer #2: No

Reviewer #3: Yes

Reviewer #4: Yes

3. Have the authors made all data underlying the findings in their manuscript fully available?

Reviewer #1: No

Reviewer #2: Yes

Reviewer #3: Yes

Reviewer #4: Yes

4. Is the manuscript presented in an intelligible fashion and written in standard English?

Reviewer #1: Yes

Reviewer #2: Yes

Reviewer #3: Yes

Reviewer #4: Yes

5. Review Comments to the Author

Reviewer #1: The authors describe some changes to an aerosol sampling procedure that appears to improve the sensitivity in detecting sputum-positive TB patients who exhale infectious aerosols. There are some serious problems with the manuscript.

Thank you for your review. The points raised are addressed below:

1) The paper compares data from the current study (RASC-2) with a previous study (RASC-1). Data from RASC-1 have already been published (Ref 14) and are repeated here.

This is true as we make clear by referencing the previous study. To our knowledge this does not contravene the ICMJE recommendations as a new research question is being addressed.

2) The two studies cannot be compared. They were conducted 2-3 years apart with different patient groups. 

We have compared baseline characteristics between the two groups of patients and found no significant differences. We are not aware of a biologically plausible reason why there would be a change in Mtb bioaerosol production in TB patients over time between the two sampling periods. 

More importantly, two different measurements were used to determine the number of mycobacteria in the sampled aerosols – viable bacilli or cfu (RASC-1) and fluorescence microscopy (RASC-2). 

The purpose of this study is to demonstrate the improvement in multiple aspects of the bioaerosol sampling methodology including both the sampling process and the method of organism detection. The following sentence in the final paragraph of the introduction has been amended to clarify this:

“Modifications to both the sampling process and the method of organism detection are described with the results of sampling from both healthy controls and individuals with newly diagnosed pulmonary TB presented to evaluate the improvements.”

A caveat has been added to the discussion section.

“A limitation of this study is that modifications to the sampling protocol and the detection method were evaluated in combination. It is therefore not possible to delineate the extent to which individual improvements impact sensitivity.”

The authors provide no evidence that the fluorescently-labeled bacilli in RASC-2 were viable.

The evidence for viability in fluorescently-labeled bacilli is referred to in ref 15. It’s beyond the scope of this study to re-examine this established dye characteristic.

3) The authors admit (lines 143-144) that the fluorescence microscopy detection assay for detecting mycobacteria in RASC-2 was subjective and dependent upon the skill of the operator. Please provide some data or references to support the validity of this assay (e.g., repeated determinations from the same sample). 

Indeed as with auramine staining in clinical practice there is a degree of operator dependence with any fluorescence microscopy detection. However, this does not invalidate the usefulness of this assay.

4) The overall conclusion of the study is trivial and can be summarized in a single sentence – “Increasing the amount of exhaled air sampled will increase the sensitivity of bacillary detection”. This is intuitively obvious to any intelligent person and does not require data or a manuscript to confirm.

We strongly disagree with this overly reductive assessment. Successfully identifying Mtb bacilli in bioaerosol is far from straight-forward and the various sampling methodologies described in the literature vary in the techniques employed and the detection yields as we have highlighted in the introduction. One aspect of the improved sensitivity is the increase in exhaled air volume sampled which we have quantified using CO2 measurements – a technique not used by other groups and not previously discussed as an explanation for low yield. Furthermore, we also describe other critical modifications such as the use of the DMN-trehalose dye for bacillus detection. This is an entirely novel approach with additional benefit in terms of a rapid time to result and assessment of organism viability. If this is a trivial we would invite the reviewer to direct us to a previous aerosol sampling study with a near 100% detection yield.

5) The authors have grossly over interpreted their results. They speculate wildly about the possible implications of their aerosol sampling results with no supporting data.

It is not clear to us what the speculative implications are to which the reviewer is referring. The only speculative statement in the discussion/conclusion is the following:

“Bioaerosol assays may offer a useful adjunct to TB transmission studies, enabling demonstration of infectivity of sputum negative individuals and the effect of specific chemotherapies on infectivity”

This is clearly not a strong (or wild) assertion but offers an indication to the reader of what the future value of this work may be. As is typical for concluding remarks, data supporting these concepts will be presented in forthcoming publications and is beyond the scope of the current paper.

6) The authors utilize jargon that is not defined for the reader. What, precisely, is an “aged” aerosol? 

The merriam-webster dictionary defines jargon as “obscure and often pretentious language marked by circumlocutions and long words”. This is not an accurate description of the term highlighted. Furthermore the intended meaning of the word “aged” is made clear in the introduction (lines 84-87)

“The guinea pig model therefore supported the concepts that infectiousness is highly heterogenous and that small bioaerosol – which arise as a consequence of dehydration of buoyant airborne droplets during “aging” following release into the environment –are primarily responsible for remote transmission”

What is the difference between “exhaled air” and “respiratory bioaerosol”?

These are basic aerobiology concepts. Respiratory bioaerosol are liquid droplets suspended in exhaled air.

Reviewer #2: In the present study, the authors have presented the method to sample bioaerosol from TB patients in order to determine the infectiousness. The method they described is an improvement of their previous work. However, I find following concerns,

Thank you for your thoughtful review. The points raised are addressed below:

1. Data for TB cases has been given for year 2018, instead of 2019.

We are not aware that global statistics have been published by the WHO with data from 2019 yet. 

2. The technical improvements in the design of RASC has not been elaborately explained. The focus has more been put on the collection outputs of aerosols in results section. I think, it will be better to present collection output in relation to modifications in the design of RASC.

We are comparing the sensitivity improvement following a package of improvements in the entire sampling and detection process i.e. comparing before and after.

“Modifications to both the sampling process and the method of organism detection are described with the results of sampling from both healthy controls and individuals with newly diagnosed pulmonary TB presented to evaluate the improvements.”

A caveat has been added to the discussion section.

“A limitation of this study is that modifications to the sampling protocol and the detection method were evaluated in combination. It is therefore not possible to delineate the extent to which individual improvements impact sensitivity.”

3. In the method section, superficial information has been given. They should describe all the modifications in detail.

Additional description has been added to the methods to 

“Briefly, the RASC is a 1.4m3 chamber in which participants can comfortably sit throughout the study period. The chamber is first sealed and then an air purge phase is performed by drawing external air across HEPA filters for a 10-minute period. The next phase is passive contamination with respiratory bioaerosol as the participant respires. A sampling phase then occurs with various devices drawing volumes of air out of the chamber and extracting the airborne bioaerosol. Finally, a second 10-minute purge phase is performed to remove residual Mtb bioaerosol.”

4. In the sample collection section, it is difficult to understand the technicalities. The authors should present these elaborately with proper diagrams and flowcharts.

An original schematic diagram of RASC-1 and a new schematic diagram of RASC-2 have been added as an appendix (referred to on line 160)

5. In fig3, statistical significance and p value are missing.

p-value has been added with the figure caption

“Fig 3. Comparison of total bacillary counts following RASC-2 sampling of sputum GeneXpert-positive TB patients (n=21) and the inter-patient empty booth RASC sampling controls (n=22). P<0.0001.”

6. Figure legends are not self explanatory, more details are required.

The figure legends have all been expanded

Reviewer #3: In the present study, authors employed modified version of Respiratory Aerosol Sampling Chamber (RASC) to incorporate liquid collection of bioaerosol using a high-flow wet-walled cyclone (RASC-2). Authors demonstrated this technical modification increased the utility of bioaerosol sampling and lowered the limit of Mtb detection in RASC-2 platform. Study result is interesting and has future application in characterizing exhaled air and bioaerosol from sputum negative and subclinical individuals. Authors have to address the following comments.

Thank you for your thoughtful and thought-provoking review. The points raised are addressed below:

Major Comments

1. Authors estimated the Mtb count per 100 liters of exhaled air as Mtb-CFU for RASC and DMN-Tre stained Mtb bacilli. Since authors employed two different methodology to estimate the lower limit of detection, study result may vary. Please clarify this discrepancy in methodology.

The purpose of this study is to demonstrate the improvement in multiple aspects of the bioaerosol sampling methodology including both the sampling process and the method of organism detection. The following sentence in the final paragraph of the introduction has been amended to clarify this:

“Modifications to both the sampling process and the method of organism detection are described with the results of sampling from both healthy controls and individuals with newly diagnosed pulmonary TB presented to evaluate the improvements.”

A caveat has been added to the discussion section.

“A limitation of this study is that modifications to the sampling protocol and the detection method were evaluated in combination. It is therefore not possible to delineate the extent to which individual improvements impact sensitivity.”

2. This includes patients with and without cavitary TB. Earlier studies have shown that pulmonary TB patients with lung cavitation are the main source of disease transmission. Did you analyze the data such as Mtb count and lower limit of Mtb detection between the patients with and without cavitary TB in both RASC-1 and RASC-2.

This is an interesting question and one which was explored in the RASC-1 study where we found no association between cavitary disease and presence of culturable aerosol. However, we feel that in the current study the number of patients is too few to further address this question and would not wish to over-interpret our results. The potential to explore the aerosol output from cavitary and non-cavitary TB does highlight the utility of our methodology and the importance of optimizing sensitivity.

3. Is it possible to study the nature of Mtb such as Mtb in the form of singles, small and large clumps in exhaled air and bioaerosol?

Again an interesting suggestion, however, we suspect this is a drawback of the cyclone sampler which may disrupt clumps of bacilli through physical agitation during sample collection.

4. There is typo in the table 1; the number of patients with lung cavitation in RASC-2 is 6 (28.6%) not 46.2%. Correct the typo.

The percentage is correct since not all had a CXR this was only the case in 13 patients. The table has been amended to make this clear.

Reviewer #4: Authors have tried to improvise on a previously established protocol for aerosol sampling from the patients for possible bacterial count. The sensitivity can improvise the testing capability for possible MTB clearance.

 Thank you for your comments

6. PLOS authors have the option to publish the peer review history of their article (what does this mean?). If published, this will include your full peer review and any attached files.

Do you want your identity to be public for this peer review? For information about this choice, including consent withdrawal, please see our Privacy Policy.

Reviewer #1: No

Reviewer #2: No

Reviewer #3: No

Reviewer #4: No

---

## [Decision Letter · Decision Letter 1]

21 Jul 2020

PONE-D-20-11317R1

Sensitivity Optimisation of Tuberculosis Bioaerosol Sampling

PLOS ONE

Dear Dr. Patterson,

Thank you for submitting your manuscript to PLOS ONE. After careful consideration, we feel that it has merit but does not fully meet PLOS ONE’s publication criteria as it currently stands. Therefore, we invite you to submit a revised version of the manuscript that addresses the points raised during the review process.

ACADEMIC EDITOR: The crux of this paper is that an improvised air sample collection system increased the sensitivity of Mtb detection. As Rev#1 mentioned in previous round of review, this could be done without comparing the two methods, since they cannot be compared head-to-head. As can be seen from the instrumentation picture, these two air collection systems are not comparable. The parameters used to collect samples, the pressure of the unit, and the Mtb detection methods are very different. As the authors noted, since multiple parameters are changed, it is hard to define which parameter contributed to the improved sensitivity. Then, a question is "why to compare these two in the first place ?". Thus, i would suggest the authors to present only the features of the latest model (RASC-2), its design, parameters and performance. The previous publication on the RASC-1 model can be discussed in the "Discussion" section, if/when relevant. Alternatively, if the authors still want to compare these two systems head-to-head, then add a table to explicitly mention all the minute differences between the two units on every parameter, such that the reader would understand what "optimization" happened from RASC-1 to become RASC-2. 

We look forward to receiving your revised manuscript.

Kind regards,

Selvakumar Subbian, Ph.D.

Academic Editor

PLOS ONE

Reviewers' comments:

Reviewer's Responses to Questions

**Comments to the Author**

1. If the authors have adequately addressed your comments raised in a previous round of review and you feel that this manuscript is now acceptable for publication, you may indicate that here to bypass the “Comments to the Author” section, enter your conflict of interest statement in the “Confidential to Editor” section, and submit your "Accept" recommendation.

Reviewer #1: (No Response)

Reviewer #2: All comments have been addressed

Reviewer #4: All comments have been addressed

2. Is the manuscript technically sound, and do the data support the conclusions?

Reviewer #1: No

Reviewer #2: Yes

Reviewer #4: Yes

3. Has the statistical analysis been performed appropriately and rigorously? 

Reviewer #1: (No Response)

Reviewer #2: Yes

Reviewer #4: Yes

4. Have the authors made all data underlying the findings in their manuscript fully available?

Reviewer #1: Yes

Reviewer #2: Yes

Reviewer #4: Yes

5. Is the manuscript presented in an intelligible fashion and written in standard English?

Reviewer #1: Yes

Reviewer #2: Yes

Reviewer #4: Yes

6. Review Comments to the Author

Reviewer #1: Please see the previous review......................................................................

Reviewer #2: (No Response)

Reviewer #4: (No Response)

7. PLOS authors have the option to publish the peer review history of their article (what does this mean?). If published, this will include your full peer review and any attached files.

Reviewer #1: No

Reviewer #2: No

Reviewer #4: No

---

## [Author Response · Author response to Decision Letter 1]

28 Jul 2020

ACADEMIC EDITOR: The crux of this paper is that an improvised air sample collection system increased the sensitivity of Mtb detection. As Rev#1 mentioned in previous round of review, this could be done without comparing the two methods, since they cannot be compared head-to-head. As can be seen from the instrumentation picture, these two air collection systems are not comparable. The parameters used to collect samples, the pressure of the unit, and the Mtb detection methods are very different. As the authors noted, since multiple parameters are changed, it is hard to define which parameter contributed to the improved sensitivity. Then, a question is "why to compare these two in the first place ?". Thus, i would suggest the authors to present only the features of the latest model (RASC-2), its design, parameters and performance. The previous publication on the RASC-1 model can be discussed in the "Discussion" section, if/when relevant. Alternatively, if the authors still want to compare these two systems head-to-head, then add a table to explicitly mention all the minute differences between the two units on every parameter, such that the reader would understand what "optimization" happened from RASC-1 to become RASC-2. 

Thank you very much for your continued assessment of this paper. 

Researchers have been trying to identify Mtb in aerosols since Chausee in 1913 with very gradual increases in the proportion of TB cases identified with Mtb positive aerosols over the last century. Our comparison of sampling and detection systems increased the proportion of patients with detectable airborne Mtb from ~40% to approaching 100% and highlights some of the specific drivers required to improve sensitivity. The lowered limit of detection is a result of the improvements in the entire system (sampler, sampling volume and detection method). Comparing sampling outputs between the two systems - in the form of exhaled air volume and bioaerosol volume (Fig. 2) – demonstrates the sampling component of this improvement. The organism recovery and detection component is demonstrated by plotting organism count per 100L of exhaled air for each patient with each system (Fig. 4). This shows the range of aerosol concentrations detected and highlights the zero-inflation problem of the less sensitive system.

We therefore feel strongly that the comparison is valid and has explanatory value. For further clarity we have added the a table to the manuscript.

---

## [Editor Report · Decision Letter 2]

12 Aug 2020

Sensitivity Optimisation of Tuberculosis Bioaerosol Sampling

PONE-D-20-11317R2

Dear Dr. Patterson,

We’re pleased to inform you that your manuscript has been judged scientifically suitable for publication and will be formally accepted for publication once it meets all outstanding technical requirements.

Kind regards,

Selvakumar Subbian, Ph.D.

Academic Editor

PLOS ONE
---

## [Editor Report · Acceptance letter]

21 Aug 2020

PONE-D-20-11317R2 

Sensitivity Optimisation of Tuberculosis Bioaerosol Sampling 

Dear Dr. Patterson:

I'm pleased to inform you that your manuscript has been deemed suitable for publication in PLOS ONE. Congratulations! Your manuscript is now with our production department. 

Kind regards, 

on behalf of

Dr. Selvakumar Subbian 

Academic Editor

PLOS ONE